# The Impact of COVID-19 on Multidrug-Resistant Bacteria at a Slovenian Tertiary Medical Center

**DOI:** 10.3390/antibiotics13030214

**Published:** 2024-02-23

**Authors:** Tatjana Mrvič, Sintija Stevanoska, Bojana Beović, Mateja Logar, Sergeja Gregorčič, Benica Žnidaršič, Katja Seme, Ivana Velimirović, Nataša Švent Kučina, Polona Maver Vodičar, Veronika Križan Hergouth, Sašo Džeroski, Mateja Pirs

**Affiliations:** 1Infection Prevention and Control Unit, University Medical Centre Ljubljana, 1000 Ljubljana, Slovenia; tatjana.mrvic@kclj.si (T.M.); mateja.logar@kclj.si (M.L.); sergeja.gregorcic@kclj.si (S.G.); benica.znidarsic@kclj.si (B.Ž.); 2Department of Infectious Diseases, University Medical Centre Ljubljana, 1000 Ljubljana, Slovenia; bojana.beovic@kclj.si; 3Jožef Stefan Institute, 1000 Ljubljana, Slovenia; sintija.stevanoska@ijs.si (S.S.); saso.dzeroski@ijs.si (S.D.); 4Department of Infectious Diseases, Faculty of Medicine, University of Ljubljana, 1000 Ljubljana, Slovenia; 5Institute of Microbiology and Immunology, Faculty of Medicine, University of Ljubljana, 1000 Ljubljana, Slovenia; katja.seme@mf.uni-lj.si (K.S.); ivana.velimirovic@mf.uni-lj.si (I.V.); natasa.svent-kucina@mf.uni-lj.si (N.Š.K.); polona.maver@mf.uni-lj.si (P.M.V.); veronika.krizan-hergouth@mf.uni-lj.si (V.K.H.)

**Keywords:** COVID-19, antimicrobial resistance, infection control, infection prevention and control measures, multidrug-resistant bacteria

## Abstract

The COVID-19 pandemic has strained healthcare systems globally. Shortages of hospital beds, reassignment of healthcare workers to COVID-19-dedicated wards, an increased workload, and evolving infection prevention and control measures have potentially contributed to the spread of multidrug-resistant bacteria (MDRB). To determine the impact of the COVID-19 pandemic at the University Medical Center Ljubljana, a tertiary teaching hospital, we analyzed the monthly incidence of select bacterial species per patient from 2018 to 2022. The analysis was performed for all isolates and for MDRB isolates. The data were analyzed separately for isolates from all clinical samples, from blood culture only, and from clinical and surveillance samples. Our findings revealed an increased incidence density of patients with *Enterococcus faecium*, *Staphylococcus aureus*, *Escherichia coli*, and *Pseudomonas aeruginosa* isolates from clinical samples during the COVID-19 period in the studied hospital. Notably, the incidence density of MDRB isolates—vancomycin-resistant *E. faecium*, extended-spectrum betalactamase-producing *K. pneumoniae*, and betalactam-resistant *P. aeruginosa*—from clinical samples increased during the COVID-19 period. There were no statistically significant differences in the incidence density of patients with blood culture MDRB isolates. We observed an increase in the overall MDRB burden (patients with MDRB isolates from both clinical and surveillance samples per 1000 patient days) in the COVID-19 period in the studied hospital for vancomycin-resistant *E. faecium*, carbapenem-resistant *K. pneumoniae,* and betalactam-resistant *P. aeruginosa* and a decrease in the methicillin-resistant *S. aureus* burden.

## 1. Introduction

The COVID-19 pandemic has profoundly impacted healthcare systems worldwide. This crisis has strained resources, marked by shortages of healthcare workers (HCWs), COVID-19-dedicated regular hospital beds, and especially intensive care unit (ICU) beds, as well as protective personal equipment (PPE) [1,2,3]. In most countries, new ICUs and hospital beds had to be opened, and HCWs were reassigned and had an increased workload due to the shortage of HCWs [1]. These challenges, coupled with the evolving nature of infection prevention and control measures (IPCMs) in the early stages of the pandemic, often contributed to diminished compliance with IPCMs, possibly exacerbating the spread of multidrug-resistant bacteria (MDRB) [4,5,6]. Several meta-analyses and reviews have shown that the impact is difficult to assess because there is considerable heterogeneity in the antimicrobial resistance (AMR) metrics used and the rate of resistance reported across various studies [4,5,6].

Our study, conducted at the University Medical Center Ljubljana (UMCL), a tertiary teaching hospital in Ljubljana, Slovenia, presents insights into the intersection of the COVID-19 crisis and the emergence or spread of MDRB. By examining the shifts in MDRB incidence during the pandemic, we aim to offer a critical perspective on infection control during a healthcare crisis and provide a comprehensive view of the MDRB landscape within our medical center’s unique context. Slovenia is located in a region with very variable AMR rates. AMR rates of invasive isolates are higher in most neighboring countries (i.e., Italy, Croatia, and Hungary), as well as in Serbia and Bosnia and Herzegovina, leading to a significant potential for cross-border spread of MDRB, making efficient IPCMs essential [7,8,9].

To evaluate the impact of the COVID-19 pandemic, we analyzed the incidence of selected bacterial species per patient, aggregated on a monthly basis, which reflects the incidence density of infections and/or colonization per 1000 patient days, hereafter referred to as the incidence density. The dataset encompasses isolates in all clinical samples, in blood cultures only, and in combined clinical and surveillance samples to assess the overall MDRB burden. We gathered data from January 2018 through to December 2022, allowing us to clearly categorize them into two timeframes for analysis: the pre–COVID-19 (pre-pandemic) period (January 2018–February 2020) and the COVID-19 (pandemic) period (March 2020–December 2022), stratified by COVID-19 and non-COVID-19 patient days.

## 2. Results

The results of the statistical analyses of the impact of the COVID-19 pandemic on the incidence of each bacterial species (*Acinetobacter baumannii*, *Escherichia coli*, *Enterococcus faecium*, *Klebsiella pneumoniae*, *Pseudomonas aeruginosa,* and *Staphylococcus aureus*) and corresponding MDRB (carbapenem-resistant *A. baumannii* (CRAb), vancomycin-resistant *E. faecium* (VRE-EFA), extended-spectrum beta-lactamase-producing *E. coli* (ESBL-EC) and *K. pneumoniae* (ESBL-KPN), carbapenem-resistant *K. pneumoniae* (CRE-KPN), beta-lactam resistant *P. aeruginosa* (CRPs-PA), methicillin-resistant and *S. aureus* (MRSA)) are shown in Table 1. CRPs-*P. aeruginosa* was defined according to Slovenian national guidelines as an isolate resistant to all classes of antipseudomonal beta-lactams [10].

Two statistical analyses were conducted, comparing the incidence density, for each bacterial species in turn, between (1) the total patient population during the pre-pandemic period and the total patient population during the pandemic period, as well as (2) the total patient population during the pre-pandemic period and the population of non-COVID-19 patients during the pandemic period. Both statistical analyses show similar results for each bacterial species.

### 2.1. Acinetobacter baumannii

There was no significant difference in the incidence density of *A. baumannii* from clinical samples between the pre-pandemic and pandemic periods (Table 1). The incidence density of patients with CRAb isolates from clinical samples per 1000 patient days is shown in Figure 1. The incidence density of patients with CRAb from clinical samples increased during the pandemic period; however, the differences were not statistically significant (Figure 1, Table 1). There was no significant difference in the incidence density of patients with blood culture CRAb isolate between the pre-pandemic and pandemic periods (Table 1).

There was also no significant difference in the overall CRAb burden (patients with CRAb isolate from both clinical and surveillance samples per 1000 patient days) between the pre-pandemic and pandemic periods (Table 1).

### 2.2. Esherichia coli

The incidence density of *E. coli* from clinical samples increased during the pandemic period, and the difference was statistically significant (*p* < 0.001; Table 1). The incidence density of patients with ESBL-EC isolates from clinical samples per 1000 patient days is shown in Figure 2; there was no significant difference in the incidence density between the pre-pandemic and pandemic periods (Table 1). There was no significant difference in the incidence density of patients with blood culture ESBL-EC isolates between the pre-pandemic and pandemic periods (Table 1).

There was no significant difference in the overall ESBL-EC burden (patients with ESBL-EC isolates from both clinical and surveillance samples per 1000 patient days) between the pre-pandemic and pandemic periods (Table 1).

### 2.3. Enterococcus faecium

The incidence density of *E. faecium* from clinical samples increased during the pandemic period, and the difference was statistically significant (*p* < 0.001; Table 1). The incidence density of patients with VRE-EFA isolates from clinical samples per 1000 patient days is shown in Figure 3; the incidence density of patients with VRE-EFA from clinical samples increased during the pandemic period, and the difference was statistically significant (*p* < 0.001; Table 1). The incidence density of patients with blood culture VRE-EFA isolates also increased during the pandemic period, and the difference was not statistically significant (Table 1).

The overall VRE-EFA burden (patients with VRE-EFA isolate from clinical and surveillance samples per 1000 patient days) increased during the pandemic period, and the difference was statistically significant (*p* < 0.001; Table 1).

### 2.4. Klebsiella pneumoniae

There was no significant difference in the incidence density of patients with *K. pneumoniae* isolates between the pre-pandemic and pandemic periods (Table 1). The incidence density of patients with ESBL-KPN isolates from clinical samples per 1000 patient days is shown in Figure 4. The incidence density increased in the pandemic period, and the difference was statistically significant (*p* < 0.001; Table 1). There was no significant difference in the incidence density of patients with blood culture ESBL-KPN isolates between the pre-pandemic and pandemic periods (Table 1).

There was no significant difference in the overall ESBL-KPN burden (patients with ESBL-KPN isolates from both clinical and surveillance samples per 1000 patient days) between the pre-pandemic and pandemic periods (Table 1).

The incidence density of patients with CRE-KPN isolates from clinical samples is shown in Figure 5. There was a statistically significant rise in the overall *K. pneumoniae* (CRE) burden (patients with CRE-KPN isolates from both clinical and surveillance samples per 1000 patient days), particularly toward the end of the period (*p* < 0.001; Table 1).

### 2.5. Pseudomonas aeruginosa

The incidence density of *P. aeruginosa* isolates from clinical samples increased during the pandemic period, and the difference was statistically significant (*p* < 0.001; Table 1). The incidence density of CRPs-PA from clinical samples per 1000 patient days is shown in Figure 6. The increase in the incidence density also increased during the pandemic period, and the difference was statistically significant (*p* < 0.001; Table 1). The incidence density of patients with blood culture CRPs-PA isolates decreased during the pandemic period, although the difference was not statistically significant (Table 1).

The overall CRPs-PA burden (patients with CRPs-PA isolates from clinical and surveillance samples per 1000 patient days) increased during the pandemic period (Figure 6), and the difference was statistically significant (*p* < 0.001; Table 1).

### 2.6. Staphylococcus aureus

The incidence density of patients with *S. aureus* isolates from clinical samples is shown in Figure 7. We found a significant (*p* < 0.001) increase during the pandemic period (Table 1). There was no significant difference in the incidence density of patients with MRSA from clinical samples between the pre-pandemic and pandemic periods (Table 1). The difference in the incidence density of patients with blood culture MRSA isolates between the pre-pandemic and pandemic periods was not statistically significant (Table 1).

The overall MRSA burden (patients with MRSA isolates from both clinical and surveillance samples per 1000 patient days) decreased during the pandemic period (Figure 7), and the difference was statistically significant (*p* < 0.001; Table 1).

## 3. Discussion

From the early days of the COVID-19 pandemic, it was clear that the impact of the pandemic on AMR and the incidence of MDRB would be difficult to predict and assess because the pandemic resulted in many significant and diverse impacts on healthcare systems. The impact of the pandemic on antimicrobial consumption and stewardship, as well as evolving IPCM guidance, particularly in the early stages of the pandemic, when IPCM recommendations were focused on the prevention of highly contagious infectious disease, particularly through the double-gloving procedure, which limited opportunities for proper hand hygiene [11] and difficulties in maintaining proper IPCMs at overburdened hospitals, may have influenced AMR rates and contributed to the spread of MDRB [12,13]. At the beginning of the pandemic, it was noted that the changes in AMR would most likely vary depending on the setting within and between hospitals and even between countries [13]. Several meta-analyses and reviews have demonstrated the difficulty in assessing the impact of the COVID-19 pandemic, with considerable heterogeneity in the AMR metrics used, the time periods and settings analyzed, and the rate of resistance reported across various studies. The consequences of the COVID-19 pandemic appear to be quite variable, with wide local variations in the impact of the COVID-19 pandemic on the AMR and transmission of MDRB in hospitals [4,5,6].

We investigated the impact of the COVID-19 pandemic on the incidence of MDRB at UMCL, a tertiary teaching hospital. At the time of writing, this is the only such study for a Slovenian hospital. Slovenia is situated in a region with very variable AMR rates. According to the European Centre for Disease Prevention and Control (ECDC), European Antimicrobial Resistance Surveillance Network (EARS-Net), and World Health Organization (WHO) Central Asian and European Surveillance of Antimicrobial Resistance (CAESAR) network data, the AMR rates of invasive isolates are higher in most neighboring countries, i.e., Italy, Croatia, and Hungary, as well as in Serbia and Bosnia and Herzegovina, leading to significant potential for the cross-border spread of MDRB, making efficient IPCMs essential to prevent local, regional, and interregional spread [7,8,9]. We analyzed trends over a 5-year period, which we divided into two timeframes: the pre-pandemic period, between January 2018 and February 2020, and the pandemic period, spanning the months from March 2020 through to December 2022. The two analyses (total pre-pandemic versus total pandemic and total pre-pandemic versus non-COVID-19 pandemic) showed almost identical results. The longer time period considered allowed us to better assess the impact of the COVID-19 pandemic on MDRB trends.

We found an overall increase in the incidence density of patients with *E. faecium*, *S. aureus*, *E. coli*, and *P. aeruginosa* isolates from clinical samples per 1000 patient days in UMCL between the pre-pandemic and pandemic periods. Analysis of the incidence density of patients with MDRB from clinical samples between the pre-pandemic and pandemic period demonstrated an increase in VRE-EFA, ESBL-KPN, and CRPs-PA from clinical samples per 1000 patient days during the pandemic period.

The COVID-19 pandemic led to significant changes in AMR trends for invasive isolates [7,8]. A recent study by the ECDC revealed a large increase in CRAb in the European Union and European Economic Area during the first two years of the pandemic [14]. The latest ECDC EARS-Net data (up to 2022) show that CRAb has increased in blood cultures in Slovenia and all neighboring countries, with the exception of Austria. Even more concerning, CRE-KPN from blood cultures also shows increasing trends in Slovenia, Croatia, and Hungary [8]. Interestingly, there were no statistically significant differences in the incidence density of patients with blood culture MDRB isolates in UMCL between the pre-pandemic and pandemic periods. Slovenian EARS-Net data show a significant increase in invasive CRAb isolates during the pandemic period, from 19.4% (34 patients) in 2020 to 66.0% (124 patients) in 2021, with a decrease to 43.3% (60 patients) in 2022 [8,14]. Although we observed outbreaks of CRAb at UMCL, our analysis shows that the proportion of invasive infections in UMCL did not increase significantly. Contrary to the data for the UMCL, Slovenian EARS-Net data also show a significant increase in invasive ESBL-KPN and CRE-KPN isolates. For other EARS-Net pathogens, the proportion of invasive isolates in Slovenia did not change significantly [8].

As expected, we observed an increase in the overall MDRB burden (defined as patients with MDRB isolates from both clinical and surveillance samples per 1000 patient-days) in the pandemic period for VRE-EFA, ESBL-KPN, and CRPs-PA and a decrease in the MRSA burden, which is consistent with our observations during the pandemic period. The increasing number of patients colonized or infected with MDRB is concerning and increases the likelihood of further spread in the future, as the majority of patients are typically colonized with MDRB 3 or more months after the initial detection of colonization [15,16].

The UMCL Infection Prevention and Control Unit (IPC) focused on maintaining IPCMs during the pandemic. The main hurdles for the UMCL IPC to control the spread of MDRB were the lack of patient rooms during COVID-19 peaks on COVID-19-dedicated wards and, in particular, the insufficient distance between adjacent beds in some of the larger wards, the lack of isolation rooms and toilets for patients with MDRB, and a large proportion of inexperienced HCWs. With the increasing number of COVID-19 cases, the capacity was ramped up with wards in seven different buildings (Figure 8). Although the management teams on COVID-19-dedicated wards included one infectious disease specialist and one nurse from the Department of Infectious Diseases (DIP), the other HCWs came from different departments. Furthermore, they were supported by nursing and medical students and even volunteers. Although all had to complete a PPE workshop, they lacked experience with IPCMs. During the first two waves, the early IPCM recommendations, focusing on the prevention of highly contagious infectious disease, may have facilitated the spread of MDRB, particularly through the double-gloving procedure, which limited opportunities for proper hand hygiene [11]. Although the IPCM guidance was quickly updated in accordance with new data and the UMCL IPC changed the recommendation on glove use accordingly [12], compliance of HCWs with new recommendations on COVID-19-dedicated wards was slow and took several months to fully implement. During this time, we observed a number of MDRB outbreaks; however, the comprehensive IPCM response of the UMCL IPC and the fact that the dedicated COVID-19 wards were located in seven different buildings helped limit the spread of most MDRB. Unfortunately, this was not the case for the spread of VRE-EFA, which has proven to be the most challenging to contain. Some of the increase in the burden may be attributed to increased surveillance due to VRE outbreaks and an increase in VRE in clinical samples. One large systematic review found a number of studies that similarly reported an increase in VRE infection and/or colonization during the pandemic in medical centers in different countries [4,17,18,19,20]. An analysis by the Centers for Disease Control and Prevention (CDC) also showed an increase in hospital-onset VRE in the United States [5]. Documented CRAb outbreaks on UMCL COVID-19 wards were observed, but they did not lead to an increase in the density of patients with CRAb from clinical samples, nor did they lead to a significant increase in the overall burden of CRAb. The incidence of CRAb in Slovenia has been variable over the years, with local hospital outbreaks and regional and interregional spread often associated with the cross-border transfer of patients followed by local spread in the past [7,21]. We also noted an increase in the number of patients colonized and infected with carbapenemase-producing *K. pneumoniae* (CPE-KPN). Although this increase was not statistically significant, the increase in the overall CRE-KPN burden was statistically significant. There were several small CPE-KPN outbreaks, which were presumably limited due to strict surveillance measures, immediate comprehensive IPCM response, and unconnected COVID-19-dedicated wards at seven different locations. Our study also demonstrated an increase in *S. aureus* clinical isolates, with no changes to MRSA bloodstream infections and a decrease in the overall MRSA burden. A large systematic review found a number of studies that reported an increase in MRSA infection/colonization during the pandemic [4,22,23], whereas another found no significant changes in the incidence density or proportion of MRSA [6]. Interestingly, an analysis by the CDC showed an increase in hospital-onset MRSA cases, with a decrease in community-acquired MRSA cases in the United States [5]. Our study did not include data on community- vs. hospital-acquired infections, so a direct comparison is not possible.

As several meta-analyses and reviews have shown, the impact of the COVID-19 pandemic is difficult to assess because there is considerable heterogeneity in the AMR metrics used, the time periods and settings analyzed, and the rate of resistance reported across various studies. They show quite variable consequences of the COVID-19 pandemic, with wide local variations in the impact of the COVID-19 pandemic on the transmission of MDRB in hospitals [4,5,6]. The large differences probably reflect different local antimicrobial consumption and stewardship, the implementation of IPCM before and during the pandemic, and the specific MDRB that were circulating in the wards when the burden due to the pandemic started to increase. As noted at the beginning of the pandemic, the full impact of the COVID-19 pandemic on AMR will only become clear over several years. The changes in AMR will also most likely vary depending on the setting within and between hospitals and even between countries [13].

Our study has some limitations. First, it is a single-center study. Second, we analysed the data at the hospital level and not at the individual patient level, which also precluded distinguishing between community-acquired and hospital-acquired infections.

In conclusion, we report on the impact of the COVID-19 pandemic on the incidence density of MDRB in UMCL, the largest hospital in Slovenia. Slovenia is located in a region with very variable AMR rates, leading to significant potential for the cross-border spread of MDRB. Our data highlight the importance of maintaining IPCM at, or close to, pre-pandemic levels during the pandemic. While the incidence density of patients with VRE-EFA, ESBL-KPN, and CRPs-PA from clinical samples increased during the COVID-19 period in the studied hospital, there were no statistically significant differences in the incidence density of patients with blood culture MDRB isolates. The increase in the overall burden of VRE-EFA, CRE-KPN, and CRPs-PA in the studied hospital is concerning, as the increasing number of colonized or infected patients increases the likelihood of further spread.

## 4. Materials and Methods

Setting: The UMCL is one of Slovenia’s two tertiary medical centers and a teaching hospital. During the study period, between 2018 and 2022, the hospital had an average of 2100 hospital beds, with 114,400 admissions per year. The number of patient days for the UMCL during the study period was stratified by COVID-19 and non-COVID-19 patients.

During COVID-19, the hospital prepared for management of COVID-19 patients by setting up dedicated wards [24]. Initially, COVID-19-dedicated wards (regular and ICU beds) were set up at the DIP (B1). With the increasing number of COVID-19 cases, the capacity was ramped up with wards in seven different buildings (B1–B7), with a maximum capacity of over 300 adult regular beds, 27 high-dependency unit beds, and 49 ICU beds (Figure 8).

The management teams on each dedicated ward included one infectious disease specialist and one DIP nurse. Other HCWs came from various departments, including surgical, gynecology, and ophthalmology wards, and so on. The teams were also supplemented with nursing and medical students. The UMCL IPC unit supported all COVID-19-dedicated wards. All HCWs reassigned to COVID-19-dedicated wards had to complete a practical workshop on the correct use of PPE before starting work. The Institute of Microbiology and Immunology (IMI) at the University of Ljubljana’s Faculty of Medicine actively notified the IPC of new MDRB cases and had weekly online meetings with the UMCL IPC. IPC personnel, who maintained pre-pandemic activities with active assessment of each individual situation in terms of patient movement prior to MDRB detection and contacts on each ward, as well as individual patient factors (patient mobility, treatment requirements, physiotherapy, etc.). A comprehensive multimodal IPC response included contact screening, patient mark-up in the hospital information system, cohorting, or transfer of patients when possible. The dedicated ward infectious disease specialist in charge was informed of the situation, and concise instructions were given to ward HCWs. IPC personnel also conducted regular audits of each individual COVID-19-dedicated ward to assess PPE use and cleaning procedures.

Microbiological data: The data on bacterial isolates and their antimicrobial susceptibility were retrieved from the laboratory information system (SRC Infonet, Naklo, Slovenia) at the IMI.

The isolates were identified using MALDI-TOF LT Microflex (Bruker Daltonics, Bremen, Germany). Antimicrobial susceptibility and resistance phenotypes were determined using EUCAST guidelines, and detailed information can be found in Appendix A [25,26]. Data were collected on the following bacteria: *A. baumannii*, *E. faecium*, *E. coli*, *K. pneumoniae*, *P. aeruginosa*, and *S. aureus*. The following MDRB were included in this study: MRSA, VRE-EFA, ESBL-EC and ESBL-KPN, CRE-KPN, CRAb, and CRPs-PA. CRPs-PA was defined according to Slovenian national guidelines as an isolate resistant to all classes of antipseudomonal beta-lactams [10].

Routine surveillance cultures were performed according to UMCL guidelines (i.e., upon admission and transfer between wards, twice a week in the ICU).

Statistical analysis: Monthly data on the first isolate of bacterial species per patient per year in a population were included in the analysis. The analysis was performed separately for patients with isolates in (i) all clinical samples, (ii) blood cultures, and (iii) clinical and surveillance samples. Data were analyzed separately for all isolates of bacterial species and for MDRB isolates, encapsulating the incidence density of infections per 1000 patient days.

The data were categorized into two timeframes: the pre-pandemic period, between January 2018 and February 2020, and the pandemic period, spanning the months from March 2020 through December 2022. Data were stratified by COVID-19 and non-COVID-19 patient days allowing two comparisons: (1) the total patient population during the pre-pandemic period versus total patient population during the pandemic period, as well as (2) the total patient population during the pre-pandemic period versus the population of non-COVID-19 patients during the pandemic period.

Variables and measures: This study examined four key questions to understand the impact of the pandemic on MDRB: (1) changes in the incidence density: this measured the overall patients with bacterial isolate from clinical samples per 1000 patient days; (2) changes in the incidence density of MDR infections: this focused on the proportion of patients with MDRB isolate from clinical samples per 1000 patient days (incidence density); (3) changes in the incidence density of invasive MDR infections: this specifically measured the proportion of patients with blood culture MDRB isolate per 1000 patient days (incidence density); and (4) changes in the incidence density of MDR burden: this measured the overall patients with MDRB isolate from both clinical and surveillance samples per 1000 patient days.

The analysis of the data that we performed is twofold. The first part includes summation and percentage differences. For each bacterium, for each question, the incidence density was summed for the two periods separately, and the results are shown as the pre-pandemic total and pandemic total (Table 1). The percentage difference between the sums provided an initial understanding of the changes in incidence density across the two periods. The second part is the Mann–Whitney *U* test [27], which was applied to the datasets. This non-parametric test helps in assessing whether the differences in the distributions of incidence density (and densities) in the pre-pandemic and pandemic periods are statistically significant. Intuitively, the test measures whether the probability of *X* > *Y* and *Y* > *X* is different, whereby *X* and *Y* are randomly chosen incidence densities from the pre-pandemic and pandemic periods, respectively. We conducted the test by using the SciPy Python library and its implementation of the Mann–Whitney *U* test. A *p*-value of less than 0.05 was considered significant, and—although the test calculates both the *U* statistic and the *p*-value—the latter was the main indicator used to interpret the statistical significance of the differences observed between the two periods.

## Figures and Tables

**Figure 1 antibiotics-13-00214-f001:**
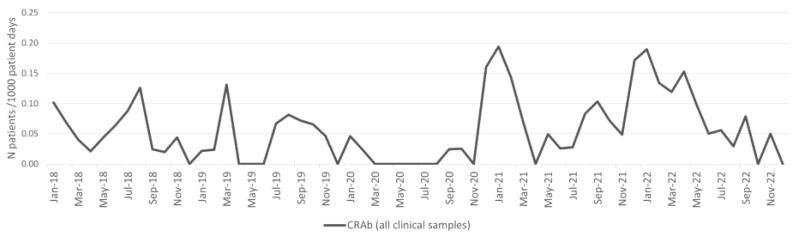
Incidence of CRAb from all clinical samples per 1000 patient days.

**Figure 2 antibiotics-13-00214-f002:**
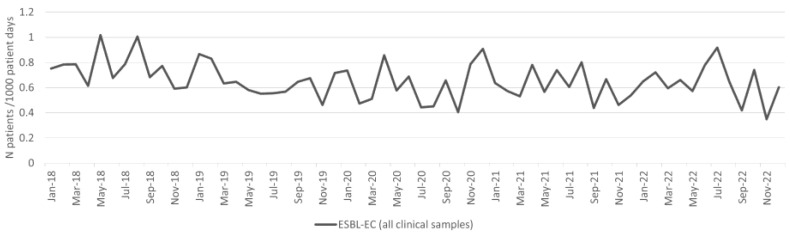
Incidence of ESBL-EC from all clinical samples per 1000 patient days.

**Figure 3 antibiotics-13-00214-f003:**
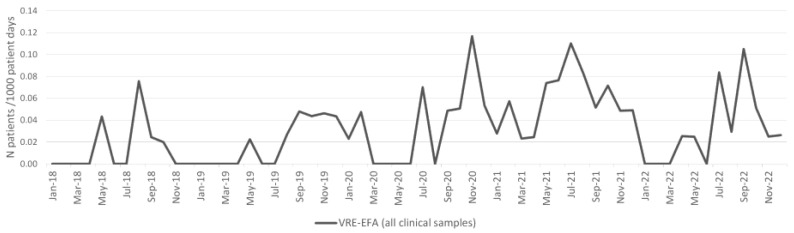
Incidence of VRE-EFA from all clinical samples per 1000 patient days.

**Figure 4 antibiotics-13-00214-f004:**
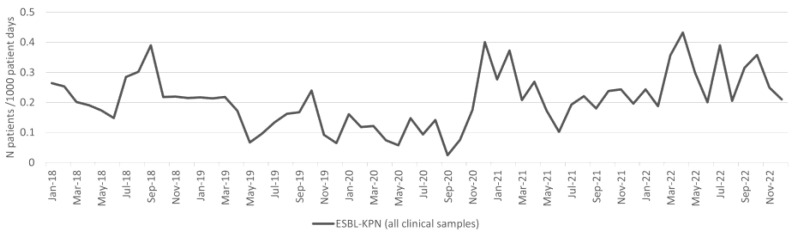
Incidence of ESBL-KPN from clinical samples per 1000 patient days.

**Figure 5 antibiotics-13-00214-f005:**
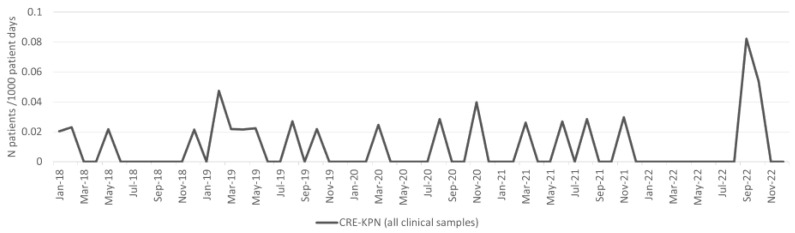
Incidence of CRE-KPN from clinical samples per 1000 patient days.

**Figure 6 antibiotics-13-00214-f006:**
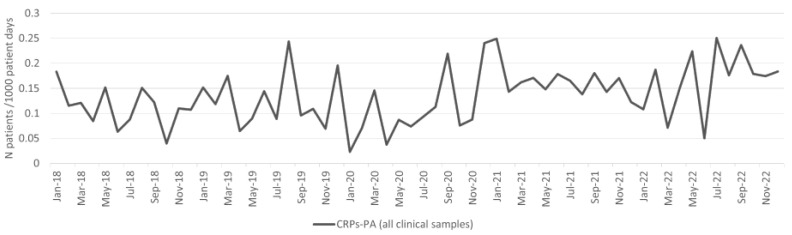
Incidence of CRPs-PA from clinical samples per 1000 patient days.

**Figure 7 antibiotics-13-00214-f007:**
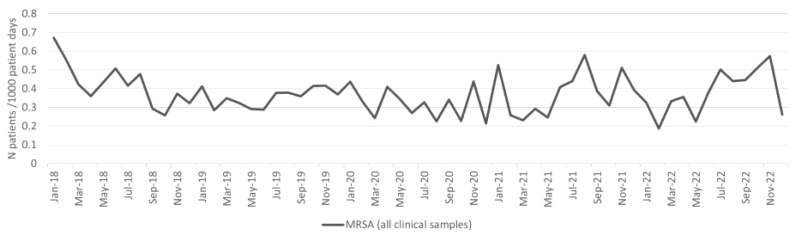
Incidence of MRSA from clinical samples per 1000 patient days.

**Figure 8 antibiotics-13-00214-f008:**
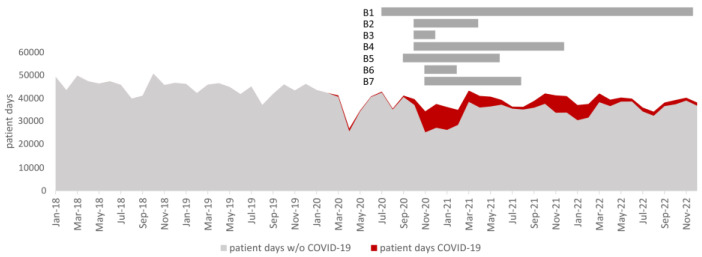
UMCL patient days between 2018 and 2022 stratified by COVID-19 and non-COVID-19 patient days. The time periods of the dedicated COVID-19 wards in seven buildings (B1–B7) are shown above the chart.

**Table 1 antibiotics-13-00214-t001:** Results of statistical analyses on the impact of the COVID-19 pandemic on the incidence of each bacterial species (*Acinetobacter baumannii*, *Escherichia coli*, *Enterococcus faecium*, *Klebsiella pneumoniae*, *Pseudomonas aeruginosa*, and *Staphylococcus aureus*) and corresponding MDRB: carbapenem-resistant *A. baumannii* (CRAb), vancomycin-resistant *E. faecium* (VRE-EFA), extended-spectrum beta-lactamase–producing *E. coli* (ESBL-EC) and *K. pneumoniae* (ESBL-KPN), carbapenem-resistant *K. pneumoniae* (CRE-KPN), beta-lactam resistant *P. aeruginosa* (CRPs-PA) and methicillin-resistant *S. aureus* (MRSA). The statistically significant results (a *p*-value of less than 0.05) are highlighted in bold within the *p*-value columns.

	Pre-Pandemic Total vs. Pandemic Total—All Patients				Pre-Pandemic Total vs. Pandemic Non-COVID-19 Patients			
Bacteria	Pre-Pandemic Total	Pandemic Total—All Patients	Percentage Change	*p*-Value	Pre-Pandemic Total	Pandemic—Non-COVID-19 Patients	Percentage Change	*p*-Value
**Changes in the incidence density of patients with bacterial isolate from clinical samples per 1000 patient days**								
*A. baumannii*	4.0	5.8	44.0%	0.698089	4.0	6.6	64.7%	0.317535
*E. coli*	242.8	335.8	38.3%	**0.024283**	242.8	372.5	53.4%	**0.000130**
*E. faecium*	30.4	51.2	68.7%	**0.000142**	30.4	57.4	89.0%	**0.000008**
*K. pneumoniae*	54.3	69.8	28.6%	0.731516	54.3	77.8	43.2%	0.141717
*P. aeruginosa*	61.2	95.9	56.8%	**0.000653**	61.2	106.6	74.2%	**0.000003**
*S. aureus*	137.7	204.0	48.2%	**0.000952**	137.7	226.0	64.2%	**0.000003**
**Changes in the incidence density of patients with MDRB isolate from all clinical samples per 1000 patient days**								
CRAb	15.5	26.3	69.3%	0.243931	15.5	30.0	93.5%	0.178420
ESBL-EC	4.3	5.7	31.3%	0.899099	4.3	6.3	45.5%	0.163077
VRE-EFA	0.9	2.7	189.4%	**0.019194**	0.9	3.0	221.4%	**0.008451**
ESBL-KPN	5.3	9.3	74.3%	**0.010516**	5.3	10.4	94.9%	**0.001176**
CRPs-PA	2.9	4.7	63.2%	**0.015988**	2.9	5.3	81.1%	**0.001869**
MRSA	4.3	5.4	24.1%	0.321187	4.3	5.9	36.9%	0.887303
**Changes in the incidence density of patients with blood culture MDRB isolate per 1000 patient days**								
CRAb	11.2	15.2	36.3%	0.814302	11.2	18.2	63.5%	0.781330
ESBL-EC	5.4	7.5	38.6%	0.737137	5.4	8.3	53.8%	0.343502
VRE-EFA	0.9	4.1	334.2%	0.230420	0.9	4.6	384.0%	0.220754
ESBL-KPN	4.6	7.4	61.9%	0.385179	4.6	8.4	84.3%	0.269986
CRPs-PA	8.4	5.4	−36.1%	0.052249	8.4	5.8	−31.3%	0.075986
MRSA	6.0	6.3	4.7%	0.215509	6.0	6.8	13.9%	0.430595
**Changes in the incidence density of patients with MDRB isolate from clinical and surveillance samples per 1000 patient days**								
CRAb	2.4	3.0	24.8%	0.590810	2.4	3.5	44.5%	0.834377
ESBL-EC	46.7	57.5	23.0%	0.172265	46.7	63.4	35.8%	0.321187
VRE-EFA	2.8	10.9	294.4%	**0.000000**	2.8	12.4	346.7%	**0.000000**
ESBL-KPN	11.9	16.9	41.6%	0.487890	11.9	18.8	57.2%	0.149995
CRE-KPN	0.9	2.2	131.9%	**0.003322**	0.9	2.4	153.0%	**0.001745**
CRPs-PA	3.9	7.4	90.4%	**0.002521**	3.9	8.2	111.8%	**0.000265**
MRSA	24.9	23.8	−4.3%	**0.000014**	24.9	26.3	5.7%	**0.001965**

## Data Availability

Data are contained within the article.

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
