# Peer review of "The Impact of COVID-19 on Multidrug-Resistant Bacteria at a Slovenian Tertiary Medical Center"

_antibiotics, 2024, doi:10.3390/antibiotics13030214_

Round 1

Reviewer 1 Report

Comments and Suggestions for Authors

Dear Authors the paper is well written and the results of this work are interesting.

I have to point out only a few minor issues:

- line 325: "CRPs was defined according to national guidelines as an isolate resistant to all classes of antipseudomonal beta-lactams" The reference for this definition links to a non-English document, can you provide a reference with a translated version of this document?

- the P of  P - value must be in italic.

Author Response

Comment

response to reviewer

line 325: "CRPs was defined according to national guidelines

as an isolate resistant to all classes of antipseudomonal betalactams"

The reference for this definition links to a non-English

document, can you provide a reference with a translated version

of this document?

The linked document is on a webpage of Slovenian antimicrobial testing committee. Unfortunately there is no official translation of the document, we did however describe Slovenian definition

the P of P - value must be in italic

Thank you for noticing - This has been corrected.

Reviewer 2 Report

Comments and Suggestions for Authors

Comments on the Quality of English Language

English very difficult to understand/incomprehensible

Author Response

Comment

Response to reviewer

English very difficult to understand/incomprehensible.

English language editor has reviewed the text.

Major comments

There is lack of novelty in the manuscript. If we go through literature, it is evident that a number of similar studies have been published on COVID-19 and AMR

A number of studies have indeed been published; however, this is the only study on the impact of the pandemic from Slovenian hospital. We have analysed the data on the hospital level reducing the potential bias when analysing subpopulations (ie. ICU patients, etc).

It is not clear which infections are included in this study? Are these nosocomial infections or community-acquired infections?

We analysed the data on the hospital level and not individual patient level and have no way of determining if the infection was community acquired or hospital acquired.

Introduction section is poor and unimpressive

This section has been expanded.

There is complete chaos in the text when comes to the critical values/terms.

In abstract: “monthly incidence of bacterial species per patient per year”

In Table 1: incidence rate, incidence density

In materials and methods line 339: ‘incidence rate of infections per 1000 patient-days’

Thank you for this comment - this has been clarified.

The data credibility is a serious issue for the manuscript. It is clearly evident from Table 1.

Here two data sets q1 and q2 are given for Acinetobacter baumannii. Q1 values are 4.0 and 5.8 whereas q2 values are 15.5 and 26.3. Does it mean that values of A. baumannii given in q1 are of non-MDR bacteria?

It must be noted that A. baumannii is usually found resistant to multiple classes of antibiotics (XDR).

Yes, the first question covers all isolates of A. baumannii (MDR and non-MDR). A. baumannii is of course intrinsically resistant to many antibiotic classes, but clinically and epidemiologically, resistance to carbapenems is the main concern.

We have removed the q1-q4 from the table and improved data presentation in the table and the methods section.

Line 70: This sentence is misleading “The results of the statistical analysis of the impact of the COVID-19 pandemic on the incidence of each bacterial species and their MDRB” What is ‘their MDRB’?

Thank you for this comment – with methods at the end of the manuscript this part was too brief. We have now added the explanation - Acinetobacter baumannii, Escherichia coli, Enterococcus faecium, Klebsiella pneumoniae, Pseudomonas aeruginosa and Staphylococcus aureus – corresponding MDRB are: carbapenem-resistant A. baumannii (CRAb), vancomycin-resistant E. faecium (VRE-EFA), extended-spectrum beta-lactamase–producing E. coli (ESBL-EC) and K. pneumoniae (ESBL-KPN), carbapenem-resistant K. pneumoniae (CRE-KPN), beta-lactam resistant P. aeruginosa (CRPs-PA) methicillin-resistant and S. aureus (MRSA).

Data analysis, its presentation and interpretation is the main weakness of the manuscript. Figures and tables are hard to comprehend (font size too small to read).

Thank you for this comment. We have reduced the data presented in the figures, used larger font and limited them just to incidence density of clinical MDRB isolates. We have improved the data presentation in the table.

Statistical analysis/significance/correlation is not properly shown in any of the table/figure.

p-values are presented in the table, we have added a short description for clarification.  We have reduced the amount of data shown in figures and limited them to incidence density of clinical MDRB isolates.

There is also lack of clarity with regard to clinical samples and blood cultures. Blood is itself a clinical sample. How they have made distinction between these two is not clear.

We analysed the data for all clinical samples (first isolate of bacterial species from a clinical sample – regardless of which was first – urine, blood, lower respiratory tract etc.) and separately for blood cultures only to analyse the incidence density of invasive MDRB isolates.

There is no mention of human ethics committee, no mention of total patients/subjects, how many clinical samples analysed etc.

We have analysed the data on the hospital level (patient-days with and without COVID-19), we did not need ethical approval as no individual patient records were accessed.

No information is given related to panel of antibiotics tested, antibiotic susceptibility testing, resistance phenotypes etc.

As was already stated in the methods section (with corresponding references) - we use EUCAST guidelines for AST and resistance mechanism detection (which defines MRSA, VRE, CRE etc.. - with exception of MDR P. aeruginosa – this is defined by the Slovenian antimicrobial testing committee). Adding complete antibiotic panels means a lot of extra text, which could be distracting. This is now included in the Supplemental file.

Minor comments

Title is not self-explanatory. It fails to provide any information about study location/area/country.

We have changed the title to include this information

The basic nomenclature rules and codes are not followed. Full name should be given at first instance of appearance of the scientific name in abstract and main text.

Thank you for noticing, this has now been corrected.

Abbreviation formats used for bacterial species are not accurate at some of the places

Thank you for noticing, this has now been corrected.

Reviewer 3 Report

Comments and Suggestions for Authors

The paper could benefit from a more detailed discussion of the potential mechanisms by which the COVID-19 pandemic may have contributed to the spread of MDRB.
In Table 1, what do q1, q2, q3, and q4 mean?
On line 78, it should say "Figure 1" instead of "Figure 2".
On line 106, it should say "Figure 2" instead of "Figure 3".

Author Response

Comment

response to reviewer

The paper could benefit from a more detailed discussion of the potential mechanisms by which the COVID-19 pandemic may have contributed to the spread of MDRB

This has been discussion has been expanded.

In Table 1, what do q1, q2, q3, and q4 mean?

This has been removed both in the table and in the methods section.

On line 78, it should say "Figure 1" instead of "Figure 2".

On line 106, it should say "Figure 2" instead of "Figure 3".

Thank you for noticing - This has been corrected.

Round 2

Reviewer 2 Report

Comments and Suggestions for Authors

I appreciate the authors for the significant improvements and revision they have undertaken to enhance the data presentation and contents in the manuscript. Title as well data tables, figures and text is much improved.

I still have one suggestion. Authors should avoid repetition of "Both statistical analyses show similar results" in the Results section. Pls check lines: 118, 133, 147, 162, 185 and 200.

Author Response

We have moved the repetitive sentence to the end of the second paragraph in the Results section as a general comment for all bacterial species.